# Lung Ultrasound: A Diagnostic Leading Tool for SARS-CoV-2 Pneumonia: A Narrative Review

**DOI:** 10.3390/diagnostics11122381

**Published:** 2021-12-17

**Authors:** Luigi Maggi, Anna Maria Biava, Silvia Fiorelli, Flaminia Coluzzi, Alberto Ricci, Monica Rocco

**Affiliations:** 1Department of Central Prevention Police, Ministry of Interior, 00198 Rome, Italy; 2Department of Medical-Surgical Sciences and Translational Medicine, Via di Grottarossa 1035, Sapienza University of Rome, 00189 Rome, Italy; annamariabiava@gmail.com (A.M.B.); silvia.fiorelli@uniroma1.it (S.F.); alberto.ricci@uniroma1.it (A.R.); monica.rocco@uniroma1.it (M.R.); 3Department Medical and Surgical Sciences and Biotechnologies, Piazzale Aldo Moro 5, Sapienza University of Rome, 00185 Rome, Italy; flaminia.coluzzi@uniroma1.it

**Keywords:** lung ultrasound, COVID-19, SARS-CoV-2, pneumonia, point of care, interstitial syndrome, chest ultrasound

## Abstract

Severe acute respiratory syndrome coronavirus 2 (SARS-CoV-2) has spread worldwide causing a global pandemic. In this context, lung ultrasound (LUS) has played an important role due to its high diagnostic sensitivity, low costs, simplicity of execution and radiation safeness. Despite computed tomography (CT) being the imaging gold standard, lung ultrasound point of care exam is essential in every situation where CT is not readily available nor applicable. The aim of our review is to highlight the considerable versatility of LUS in diagnosis, framing the therapeutic route and follow-up for SARS-CoV-2 interstitial syndrome.

## 1. Introduction

At the end of 2019, a new coronavirus called SARS-CoV-2, emerged in China, which was first caused an epidemic illness in Wuhan and then spread worldwide causing a global pandemic [1]. SARS-CoV-2 has high affinity for upper and lower respiratory tract illnesses [2].

Clinical presentation of SARS-CoV-2 infection has an extremely heterogeneous spectrum of severity ranging from self-limiting infection to acute respiratory failure [3].

The diagnostic gold standard of coronavirus lung involvement is chest tomography with a 97% sensitivity to detect SARS-CoV-2 pneumonia [4].

Typical CT findings of COVID-19 related to pneumonia are bilateral involvement in 100% of cases, with evidence of ground glass opacities, crazy paving signs, subpleural lines and fat vessel signs, which involve all lung lobes in 88% of cases. CT findings suggestive of SARS-CoV-2 pneumonia correlate with abnormalities seen on chest ultrasounds [5]. Contrarily, chest X-rays have a poor ability to detect SARS-CoV-2 lung lesions [6] compared to CT and chest ultrasounds [7].

Although CT scan is the imaging gold standard, it is expensive and includes some disadvantages such as the need to move patients, risk of spreading infections and use of human resources [8].

Ultrasound assessment for COVID-19 pneumonia to date has been clear since the beginning of the pandemic [3,9,10]. LUS was also used during the previous H1N1 viral pandemic, showing a good sensitivity and specificity [11]. Compared to CT, lung ultrasound is cost effective, radiation free, repeatable and performable bedside by a single operator [12].

In the context of the SARS-CoV-2 pandemic, the high sensitivity of thoracic ultrasound and its manageability should have numerous applications; however, has been extensively used mostly in the emergency department (3) and intensive care unit (ICU) [13]. It could also be a valuable tool in the pre-hospital phase to identify those SARS-CoV-2 patients who are at high risk of developing respiratory failure requiring hospitalization.

Furthermore, it allows clinicians to monitor the long-term COVID-19 patients [14].

About 14% of SARS-CoV-2 positive patients require hospitalization and oxygen support and 5% require intensive care. Early identification of severe pneumonia allows for a more appropriate therapeutic approach to be initiated promptly in a designated hospital setting or intensive care unit. For this purpose, chest imaging (radiography, CT or LUS) is essential [15]. As compared to HRCT, LUS has the ability to detect interstitial syndrome with a sensitivity of 100% and a specificity of 97% [16]. LUS can be used in predicting outcome of SARS-CoV-2 pneumonia. It could guide the choice of different ventilatory strategies, from high flow nasal cannula to mechanical ventilation. Finally, LUS has a high predictive power of severity and fatality of SARS-CoV-2 pneumonia [17].

The role of chest ultrasound in the management of critically ill patient with respiratory symptoms has been extensively demonstrated, showing its diagnostic value, higher than chest X-ray and similar to chest CT scan in a wide range of pulmonary diseases [18]. LUS is particularly accurate in diagnosing dyspnea, differentiating between “wet” causes (cardiogenic pulmonary edema and acute inflammation) and “dry” causes (chronic obstructive pulmonary disease and asthma) [19].

Currently in this pandemic, LUS examination allows a complete evaluation of the patient with COVID-19 related interstitial pneumonia [20].

However, in this paper we underline the clear definition of the clinical context, and positive swab test in the management of the SARS-CoV-2 patient to define possible differential diagnosis of interstitial patterns as acute respiratory distress syndrome (ARDS) [21], pulmonary edema [22] and contusion [23].

## 2. Lung Ultrasound in COVID-19 Pneumonia: Technique

LUS for COVID-19 can be performed by most ultrasound machines available. A lung preset is available for many recent machines but operators can adjust settings to provide good quality images [18].

The feasibility of a lung scan does not provide a predefined scheme for the evaluation and can be adapted to the needs of the clinician.

However, we suggest depth should be set between 8 and 10 cm and modulated according to the height of the patient and the thickness of his chest wall. 

Focus should be positioned on the pleural line when available. It might be necessary to adjust the post processing settings to acquire the best image.

The ultrasound chest exploration should be systematic, starting from the anterior to the inferior areas along the intercostal spaces from medial to lateral.

If the patient is in the supine position, it is advisable to scan the anterior and lateral thorax dividing each hemithorax into 6 zones with diagnostic accuracy comparable to the 12-zone scanning scheme [24]. When the patient is in forced supine position, it may also be convenient to use an 8-zone scan scheme. In this approach, 4 zones, 2 anterior and 2 lateral, are examined for each hemithorax. The anterior zones are between the mid-clavicular line medially and the anterior axillary line laterally, while the lateral zones are between the anterior axillary line medially and the posterior axillary line laterally [25].

Regarding the ultrasound analysis of posterior regions of the thorax, which can be scanned if the patient is able to maintain the sitting position, the detection is performed along the paravertebral line, the scapular line and the posterior axillary line [26].

The ultrasound probe can be positioned transversely along the intercostal space or longitudinally perpendicular to the ribs. This last approach makes it possible to identify “the bat sign” (Figure 1), in which the bat’s wings are represented by the upper and lower ribs while the outline of the bat’s body is the pleural line [27].

It is possible to choose any transducer: cardiac, convex, micro convex or linear. Usually, the convex probe is the most used because it allows a better visualization of the pleural line and subpleural space. Moreover, it guarantees a better evaluation of the diaphragmatic recess. Alternatively, using a phased array for both cardiac and lung ultrasound can reduce costs [27].

## 3. Lung Ultrasound in COVID-19 Pneumonia: Findings

LUS Assessment for SARS-CoV-2 Pneumonia is a Clinically Driven “Point of Care” Method.

LUS detects changes in the relationship between air and tissue at the surface of the lung [12].

Ultrasound cannot be transmitted through air and allows for air-filled lungs to create artifacts.

In a normally aerated lung, a pleural line appears as a hyperechoic horizontal line that moves synchronously with breaths [28]. Moreover, we recognize several hyperechoic lines parallel to the pleura named A lines. Those are repetition artifact that indicates a normally aerated parenchyma [29]. However, in patients with chronic obstructive pulmonary disease (COPD), atelectasis of the lung and asthma, the A lines are present and well represented in the presence of normal lung sliding, while in patients with pneumothorax, the only visible artifacts are the A lines in the absence of lung sliding [25]. The Z lines are also non-pathological vertical hyperechoic artifacts that originate from the pleural line but do not reach the border of the screen, do not move with the lung sliding and do not cancel the A lines. These features allow to differentiate the Z lines from the B lines, which are also hyperechoic lines and instead can take on pathological significance.

SARS-CoV-2 pneumonia is an interstitial pneumonia with a typically peripheral distribution.

It is characterized by progressive reduction of air-filled lungs and LUS acts as a densitometer. It detects changes in the abnormal ventilated parenchyma due to lung density increasing and air content decreasing [30].

The sonographic findings suggestive of SARS-CoV-2 pneumonia are B-line, fuse B-line (white lung), abnormalities of pleural line, small and large peripherical consolidations with or without bronchogram [31].

B lines are vertical hyperechoic artifacts originating from the pleural line and extending to the bottom of the image erasing the A lines [3]. 

B line move synchronously with lung sliding and are indicative of interstitial syndrome [32]. Cluster of B- lines are the ultrasound sign of the subpleural interlobular thickening. In the scanned fields, more than three B lines or their confluence, configuring the “white lung”, suggests an interstitial pneumonia SARS-CoV-2 and the number of B lines is associated with a greater severity of pulmonary involvement [33] (Figure 2).

The Kerley B-lines visible on chest X-ray, which are the expression of the thickening of the interlobular septa in the interstitial syndrome, correlate with the ultrasound finding of B-lines in numbers greater than 3 per field [25]. In the initial phase, B lines have a focal distribution and there is a separation between them. As the disease progresses, B lines tend to merge and their distribution increases. In the resolution phase, they gradually disappear [26]. Consequently, ultrasound allows us to identify the different evolutionary stages of SARS-CoV-2 pneumonia [34].

The white lung is a multiple coalescent B line that completely occupies the lung field. It is due to alveolar de-aeration [3] and it correlates with ground glass on HRCT in SARS-CoV-2 pneumonia [3,33] (Figure 3).

Among vertical artifacts of SARS-CoV-2 pneumonia, “the light beam”, a bright vertical artifact that moves rapidly with sliding, correlates with the early phase “ground glass” observed on chest CT scan. It arises from a normal pleural line, goes within areas of normal pattern or with separate B lines and disappears quickly from the screen with an “on-off” effect [35].

Another suggestive ultrasound sign is abnormalities of the pleural line [36] (Figure 4). In the initial phase of the disease, small and diffuse irregular thickening of the pleural line appears. This artifact becomes more nodular in the appearance as the disease progresses, with areas of discontinuity that usually disappear if the disease progresses favorably [26].

In the subpleural space, consolidations can be found and appear on the ultrasound as areas of hepatization with irregular edges and air bronchogram [32].

Parenchymal consolidations increase with disease severity [37], while large consolidations with air bronchogram in the lower lobes raise the suspicion of bacterial superinfection [35] (Figure 5).

The pulmonary lesions are preferentially located in the posterior zone with bilateral distribution [38], while in severe forms it could progress to affect all lung fields [36], and have a patchy bilateral distribution of multiform cluster with sparing areas [39].

Complications of severe forms of SARS-CoV-2 pneumonia are pneumomediastinum and pneumothorax eventually associated with subcutaneous emphysema [40].

If pneumothorax (PNX) is present, there is no B line, lung pulse and sliding. If these absences are highlighted on the ultrasound scan, it is necessary to perform another scan along the middle axillary line. If the lung point is visible, the diagnosis of PNX will be probable with a sensitivity between 75% and 100% and a specificity between 94% and 100% [41].

If subcutaneous emphysema is present, on chest ultrasound it will be possible to visualize only B lines arising from the subcutaneous tissue and not from the pleural line that will not be visible as well as the subpleural space.

### Scores

Many scores have been evaluated to assess thoracic ultrasound but the most commonly used is the lung ultrasound score (LUSS).

This score is widely used to assess patients in several clinical contexts. It allows evaluation of loss of aeration in the scanned area with a numeric outcome [42]. LUSS numerically describes the spread and progressive severity of pulmonary involvement. Moreover, numerical value is reduced in the successfully extubated patient. The 12 scanning zones are evaluated and a score from 0 to 3 is assigned for each one: 0 is assigned to a normal ultrasound pattern, 1 to the presence of the B lines, 2 to the white lung and 3 to consolidations [43]. The total score goes from 0 to 36 and it correlates with increasing lung involvement severity.

Correlation between LUSS and lung weight has been extensively demonstrated [44] and it could be a useful tool for assessing severity of SARS-CoV-2 pneumonia and monitoring the progression of lung involvement [45].

LUSS evaluation for SARS-CoV-2 has been shown to be a valuable choice for ICU patients [43,45]. Furthermore SARS-CoV-2 pneumonia and SARS-CoV-2 ARDS have specific features. In fact, posterior consolidations are preponderant in SARS-CoV-2 pneumonia and this feature weakens LUSS accuracy [46].

A specific ultrasound approach for SARS-CoV-2 pneumonia has been proposed since 2020 [47].

This new score evaluates the scan of seven areas in each hemithorax (three posterior, two lateral, two anterior). The scanned are located: -On the paravertebral line upon the curtain sign.-On the para-vertebral line at the inferior angle of the scapula.-On the para-vertebral line at the spine of the scapula.-On the mid-axillary line below the inter-nipple line.-On the mid-axillary line above the inter-nipple line.-On the mid-clavicular line below the inter-nipple line.-On the mid-clavicular line above the inter-nipple line. 

This proposal needs two skilled operators and pocket devices to perform ultrasound. The aim is to minimize risk to health workers operators and reduce the ultrasound operator-dependance. The limits of this score are shortage of skilled operators and the small number of pocket devices readily available.

## 4. Application Setting

In the context of the SARS-CoV-2 pandemic, the high sensitivity of thoracic ultrasound and its manageability has numerous application settings.

These settings include the triage of patients with flu symptoms in the emergency department, monitoring of hospitalized patients in both low-intensity care wards and ICU and in the follow-up of long COVID-19 patients [14,17,48]. It could also be a valuable tool in the pre-hospital phase to identify those SARS-CoV-2 positive patients who are at high risk of developing respiratory failure requiring hospitalization.

Chest ultrasound is a non-invasive, patient-safe (radiation free) exam that can quickly rule out COVID-19 pneumonia with good diagnostic accuracy, making it particularly useful for triage of the symptomatic patient with suspicion of COVID-19 infection [3].

It has been shown that there is a correlation between ultrasound results, clinical severity, and the trend over time [36]. In the hospital, patients who need and therefore receive respiratory support must be carefully monitored in order to identify early clinical worsening and thus well timed treatment/escalation [37].

### 4.1. Emergency Department

The SARS-CoV-2 pandemic has revolutionized the medical approach to patients arriving in the emergency room with respiratory symptoms. In the current context, it is essential while awaiting swab test results, to define the severity and consequently to refer the patient to the intensive care unit or intermediate ward as needed. Lung ultrasound is therefore a first level examination that can be quickly performed bedside and that provides information about the severity of lung involvement. 

Furthermore, in the event a patient is in a critical condition that requires life-saving interventions, lung ultrasound together with blood gas analysis allows the assessment of severity without wasting precious time to move the patient to perform chest CT scan, which can be performed once the patient is stabilized. On the other hand, some authors have highlighted the predictive value of LUS to detect worsening in patients, admitted to the emergency department without severe symptoms of SARS-CoV-2 [49]. In confirmed positive patients with mild–moderate symptoms, lung ultrasound allows an initial evaluation, which together with anamnesis, stable clinical condition, blood gas analysis and chest CT, defines the appropriate care management of the patient with possible admission to an intermediate ward or discharge at home with the recommendation of isolation [3].

### 4.2. Intensive Care Unit

In an ICU setting, lung ultrasound, due to the simplicity of execution and prompt availability, is particularly useful in documenting any improvement or worsening without the need to move the patient. Bedside evaluation reduces the risk associated with transporting the unstable patient to CT, the infectious risk for healthcare personnel and possible spread of the virus during transports. Consequently, ultrasound evaluation helps in the saving of human and economic resources [50].

LUS is also known as a valid tool to optimize ventilatory therapy and its use as a guide for recruitment maneuvers [35,51]. 

Furthermore, thoracic ultrasound can be a valuable complementary tool in assessing the need for pronation as it is a good predictor of oxygenation response in the prone position as has been demonstrated in ARDS patients with a normal LUS pattern in anterior and basal zones bilaterally [52].

It is also a useful tool for monitoring regional aeration changes during prone ventilation [53]; unfortunately, lung ultrasound cannot detect over-distention (51).

Weaning mechanical ventilation is a crucial point in intensive care unit patient treatment. In this context, it is essential to have diagnostic techniques to verify ventilator weaning and ultrasounds are a tool to monitor the cardiorespiratory system during weaning, and in particular, the weakness of the respiratory muscles, especially the diaphragm, which can contribute to weaning failure [54].

Lung ultrasound is also useful in predicting post-extubation distress as it is able to assess loss of regional lung aeration during a spontaneous breathing trial [55].

### 4.3. Pediatric Patients

The pediatric population, particularly children under 6 months, are more susceptible to infections than adults and the mortality rate from seasonal flu is much higher. However, SARS-CoV-2 infection appears to be less severe in children than adults, presenting more frequently in mild or moderate form, unlike SARS-CoV-2 infection, which occurs in more severe form [56]. However, it has recently emerged a new spectrum of SARS-CoV-2 disease characterized by multi-organ inflammation and cardiovascular compromise among children, although respiratory failure is currently still the main cause of admission to the pediatric intensive care unit [57].

Radiation exposure, the requirement of sedation and infectious risk reduction are the main advantages in using lung ultrasound in pediatric population [58].

In children, half of all patients have neither symptoms nor radiological findings [59]; moreover, chest ultrasound has a greater sensitivity than chest X-ray in detecting pneumonia and avoids exposure to ionizing radiation [27]. 

In children with SARS-CoV-2 pneumonia, CT scan is mainly used to define the severity disease [59]. Pediatric patients have an increased risk to radiation exposure and often requires sedation during CT scan.

LUS findings in the pediatric age are similar to those found in adulthood and include pleural line abnormalities, B line, white lung and subpleural consolidations. LUS could be an integrative tool complementary to the CT scan in the diagnosis and monitoring of SARS-CoV-2 pneumonia in children [58]. Nevertheless, current literature in children still considers diagnosis mainly based on typical findings obtained with CT scan, epidemiology and contact tracing [56]. 

### 4.4. Out-Patients

In hospitalized patients with SARS-CoV-2 pneumonia, LUS plays an important role in care management. 

However, some authors suggest LUS utilization, performed with pocket/portable sized ultrasound, in the out-patient therapeutic route, management and follow-up [3].

In mild–moderate forms of SARS-CoV-2 pneumonia that do not require hospitalization, it is essential to monitor the evolution of the disease and promptly identify worsening of symptoms.

In home management of COVID-19 pneumonia, use of imaging techniques such as CT and chest X-ray is not readily available. In this setting, lung ultrasound, due to its high sensitivity, ease of execution at the patient’s home and cost effectiveness, can be used as a screening method for pulmonary involvement [33,60] and could have a potential role in early hospitalization. Further study should clarify it.

Moreover, is known that patients with severe SARS-CoV-2 pneumonia often have sequelae that needs follow-up and pulmonary rehabilitation [61]. Muscle impairment could have a potential role [62]. 

Some authors suggest an early follow up assessment four to six week after discharge and a further follow up at twelve weeks in severe SARS-CoV-2 pneumonia. In this route the recommended imaging method is Chest-X-ray [61].

In our knowledge in Long Term Covid patients LUS could be a potential perfect methodology to follow those patients due to cost effectiveness and radiation safeness.

## 5. Conclusions

During this pandemic, point of care lung ultrasound has demonstrated a key role in the management of patients with COVID-19 associated lung injury.

LUS provides supplemental imaging information. It shows cost effectiveness, radiation safety, reduction of the infectious risk of healthcare personnel as well as savings in human resources. It is an examination that can be performed at the patient’s bedside by a single operator, easily repeated at any time and shows a good correlation with HRCT and is better suited than the chest X-ray.

Its high sensitivity and specificity, together with its flexibility, allows it to be used for the triage of patients with flu symptoms and the monitoring of hospitalized patients in order to identify early clinical worsening and thus timely treatment.

LUS can be integrated into the diagnostic and therapeutic pathway of COVID-19 pneumonia by informing the initiation of respiratory support as well as escalation, titration and weaning from mechanical ventilation. Further study should be considered to better explain the potential role of LUS for early admission to hospital and in follow-up.

## Figures and Tables

**Figure 1 diagnostics-11-02381-f001:**
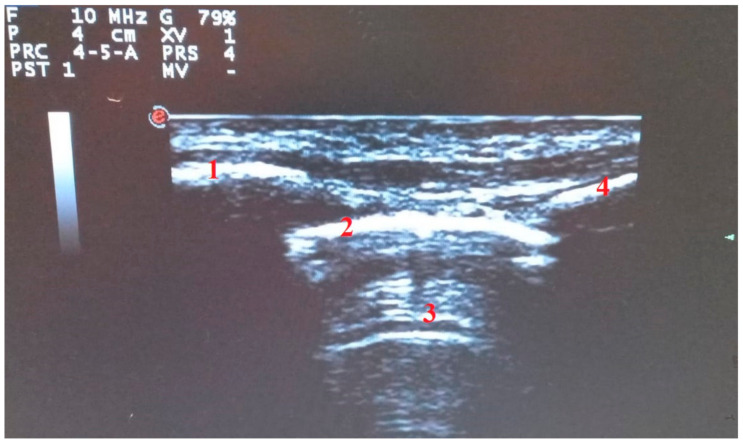
Longitudinal scan with linear probe: “the bat sign”. (1) Upper rib. (2) Pleural line. (3) A Lines. (4) Lower rib.

**Figure 2 diagnostics-11-02381-f002:**
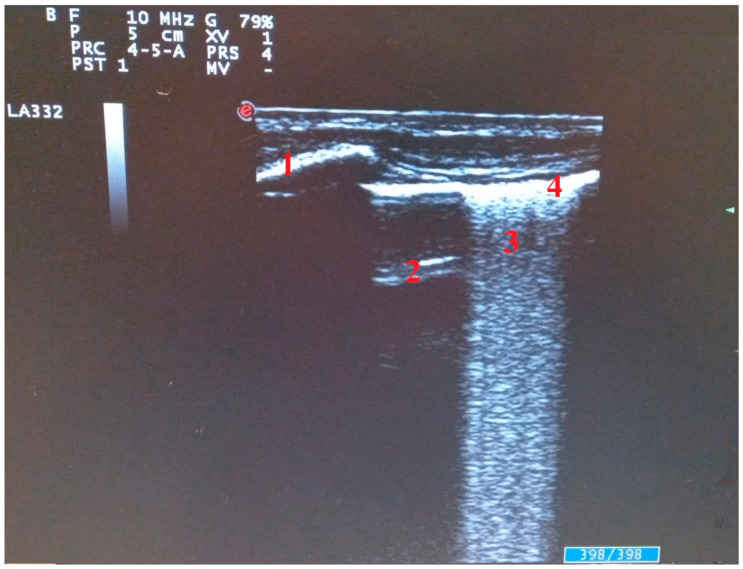
Ultrasound findings in SARS-CoV-2 pneumonia: (1) ribs, (2) A line, (3) cluster of B lines, (4) pleural line.

**Figure 3 diagnostics-11-02381-f003:**
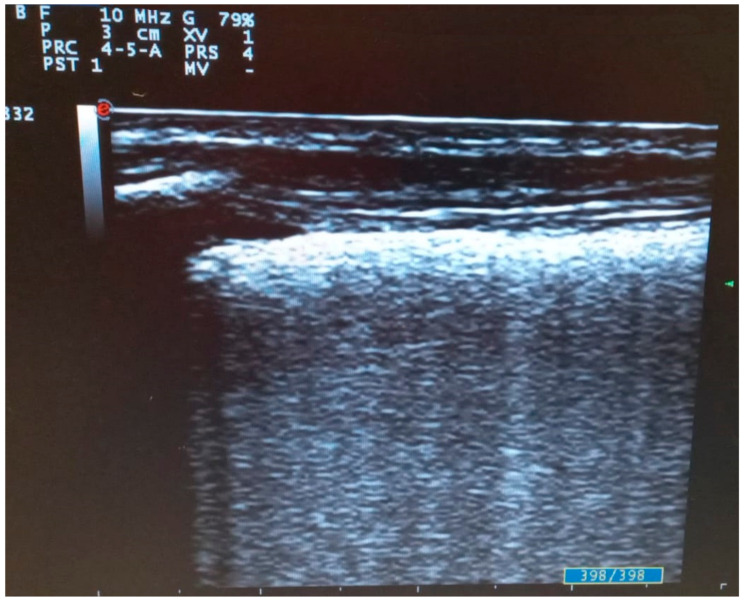
Transversal scan with linear probe of COVID-19 pneumonia: fused B lines configuring “white lung”.

**Figure 4 diagnostics-11-02381-f004:**
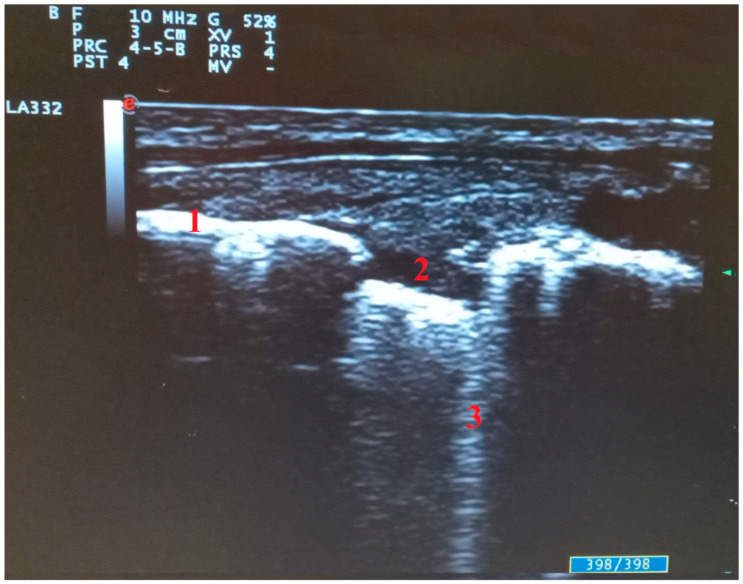
Abnormalities of pleural line in transversal scan: (1) pleural line, (2) pleural line interruption with subpleural consolidation, (3) single B line arising from subpleural consolidation.

**Figure 5 diagnostics-11-02381-f005:**
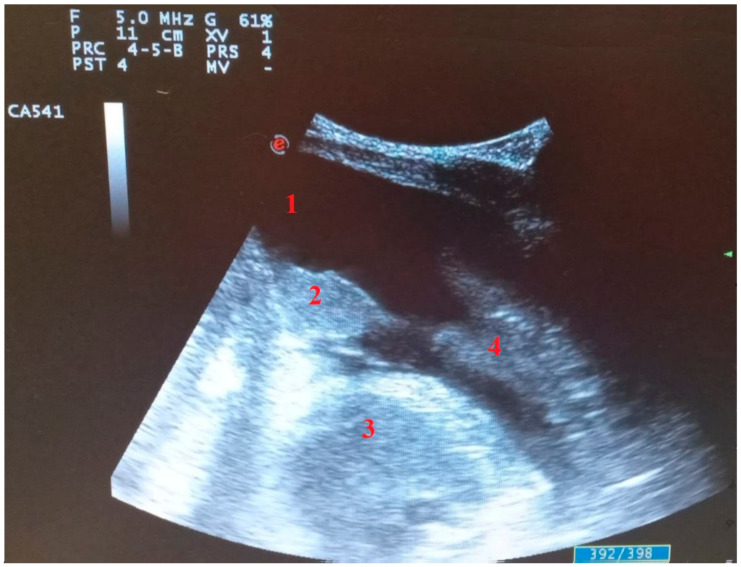
Longitudinal scan with convex probe in COVID-19 patient on mechanical ventilation with bacterial superinfection: (1) pleural effusion, (2) parenchymal consolidation without air bronchogram, (3) heart, (4) parenchymal consolidation with air bronchogram.

## Data Availability

Not applicable.

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
