# Peer review of "Lung Ultrasound: A Diagnostic Leading Tool for SARS-CoV-2 Pneumonia: A Narrative Review"

_diagnostics, 2021, doi:10.3390/diagnostics11122381_

Round 1
Reviewer 1 Report
The authors wrote a very good narrative review describing the usefulness of lung ultrasound and its scoring system in diagnosis, monitoring treatment, detecting outcomes, and follow-up of patients with COVID19 pneumonia. A broad topic on lung ultrasound is described, the choice of literature is good, and the evaluation and synthesis of evidence is clearly and prudently presented.
My review focuses only on suggestions for minor improvements in the text, and for the purpose of better presentability and completeness.
Minor revisions:
- In the introduction section, when describing the usefulness of lung ultrasound in clinical practice, I believe it should be added that lung ultrasound can also be used in predicting the outcome of covid pneumonia treatment. Specifically in the prediction of the use of higher modalities of ventilation (such as high flow nasal cannula or mechanical ventilation) as well as in prediction of a potentially fatal outcome as described in Skopljanac et al. Role of Lung Ultrasound in Predicting Clinical Severity and Fatality in COVID-19 Pneumonia. J Pers Med. 2021 Jul 30;11(8):757. doi: 10.3390/jpm11080757. PMID: 34442401; PMCID: PMC8399683.
- The acronyms and abbreviations should be addressed when firstly mentioned in the text (e.g. ARDS, PNX).
- The other thing about the definition of the acronyms is on Line 131 – HRTC – should it be HRCT?
- On first mention of the B-lines around Ln 103 I think it should be pointed out that those are “hyperechoic” lines to ease the following of the images.
- Ln 211, previously mentioned reference of Skopljanac et Al confirms this statement.
- Ln 246 – consider stating “lung recruitment maneuvers”
- Ln 254 – brackets typo
- Ln 264 – SARS COV missing 2
- Lines 285, 298 – decapitalize Authors
Author Response
Dear Reviewer 1.We want to thanks you for the advices. We changed manuscript based on your suggestions:
1) Done as requested
2) Done as requested
3) Done as requested
4) Done as requested
5) Done as requested
6) Done as requested
7) Done as requested
8) Done as requested
9) Done as requested
Reviewer 2 Report
The work presented for review concerns a very practical and currently important issue. What draws attention, however, is the undesirable content of individual paragraphs. I suggest the authors review the manuscript again and analyze its structure, organize it. I also present some major remarks below
110-112
“The sonographic findings suggestive of SARS COV 2 pneumonia are B-Line, fuse B- 110 Line (white lung), abnormalities of pleural line, light beam, consolidation with or without 111 bronchogram (27).”
This part is confusing. That is the difference between B-line add light beam?
Moreover – small consolidations are also often find in COVID-19 patients, as far as oval or triangular consolidation with vascular sign – as in peripheral PE.
The part 3 seems chaotic to me. The interweaving of the description of correct and incorrect ultrasound images seems to me to be disordered. I would propose a description of the correct ultrasound image as an introduction, and then a description of the pathological finds with classification to the stage of the disease
226-227
“If chest scan shows bilateral A pattern, it is possible to reasonably exclude SARS COV 226 2 pneumonia and as soon as the swab is confirmed negative, direct the patient to a non- 227 COVID ward.”
Is this an opinion of the outlets or is it a statement supported by research? Personally, from my own experience, I absolutely cannot agree with this. In the first phase of COVID-19, LUS is often absolutely normal, and in the case of clinical symptoms, one PCR test may not be sufficient to rule out COVID-19.
Conclusions:
“Clinician need to be aware that high sensibility and specificity is related to the “point 308 of care” analysis to avoid possible misdiagnosis and mistreatment.”
This conclusion is not entirely consistent with the text as a whole. After all, describing specific diagnostic conclusions and therapeutic implications ...
Bearing in mind the dynamics of the pandemic, it is worth considering the most recent works, there are no publications from 2021 in the literature. I suggest the authors to including at least the following
Clin. Med.2021, 10(15), 3255; https://doi.org/10.3390/jcm10153255
Diagnostics 2021, 11(1), 82; https://doi.org/10.3390/diagnostics11010082
Clin. Med.2021, 10(6), 1288; https://doi.org/10.3390/jcm10061288
Diagnostics 2021, 11(5), 761; https://doi.org/10.3390/diagnostics11050761
Author Response
Dear Reviewer 2 we wish to thank you for the advices. We changed our paper based on your suggestion.
1)“The sonographic findings suggestive of SARS COV 2 pneumonia are B-Line, fuse B- 110 Line (white lung), abnormalities of pleural line, light beam, consolidation with or without 111 bronchogram (27).”
This part is confusing. That is the difference between B-line add light beam?
Changed as requested
Moreover – small consolidations are also often find in COVID-19 patients, as far as oval or triangular consolidation with vascular sign – as in peripheral PE.
2) The part 3 seems chaotic to me. The interweaving of the description of correct and incorrect ultrasound images seems to me to be disordered. I would propose a description of the correct ultrasound image as an introduction, and then a description of the pathological finds with classification to the stage of the disease
We changed the entire Findings section as requested
3) “If chest scan shows bilateral A pattern, it is possible to reasonably exclude SARS COV 226 2 pneumonia and as soon as the swab is confirmed negative, direct the patient to a non- 227 COVID ward.”
Is this an opinion of the outlets or is it a statement supported by research? Personally, from my own experience, I absolutely cannot agree with this. In the first phase of COVID-19, LUS is often absolutely normal, and in the case of clinical symptoms, one PCR test may not be sufficient to rule out COVID-19.
Changed as requested
“Clinician need to be aware that high sensibility and specificity is related to the “point 308 of care” analysis to avoid possible misdiagnosis and mistreatment.”
This conclusion is not entirely consistent with the text as a whole. After all, describing specific diagnostic conclusions and therapeutic implications ...
Conclusion has been entirely changed as requested
Bearing in mind the dynamics of the pandemic, it is worth considering the most recent works, there are no publications from 2021 in the literature. I suggest the authors to including at least the following
Clin. Med.2021, 10(15), 3255; https://doi.org/10.3390/jcm10153255
Diagnostics 2021, 11(1), 82; https://doi.org/10.3390/diagnostics11010082
Clin. Med.2021, 10(6), 1288; https://doi.org/10.3390/jcm10061288
Diagnostics 2021, 11(5), 761; https://doi.org/10.3390/diagnostics11050761
All articles has been integrated in the body of the article.
What draws attention, however, is the undesirable content of individual paragraphs. I suggest the authors review the manuscript again and analyze its structure, organize it.
We reviewed the body of the article as requested. We organized the text in individual paragraphs based on of the editor’s suggestion in first review. If it is desiderable to re-organize without the division in paraghraph we would apply it.
Round 2
Reviewer 2 Report
Dear Authors
Thank you for considering my suggestions. Congratulations on your interesting study